# Altered Mitochondrial Function and Accelerated Aging Phenotype in Neural Stem Cells Derived from Dnm1l Knockout Embryonic Stem Cells

**DOI:** 10.3390/ijms241814291

**Published:** 2023-09-19

**Authors:** Seung-Bin Na, Bong-Jong Seo, Tae-Kyung Hong, Seung-Yeon Oh, Yean-Ju Hong, Jae-Hoon Song, Sang-Jun Uhm, Kwonho Hong, Jeong-Tae Do

**Affiliations:** 1Department of Stem Cell and Regenerative Biotechnology, Konkuk Institute of Technology, Konkuk University, 120 Neungdong-ro, Gwangjin-gu, Seoul 05029, Republic of Korea; amanda1220@naver.com (S.-B.N.); sbj1990@naver.com (B.-J.S.); htk0518@naver.com (T.-K.H.); osy3140@gmail.com (S.-Y.O.); ndhong7@gmail.com (Y.-J.H.); picpoket15@naver.com (J.-H.S.); hongk@konkuk.ac.kr (K.H.); 2Department of Animal Science, Sangji University, Wonju 26339, Republic of Korea; sjuhm@sangji.ac.kr

**Keywords:** *Dnm1l*, mitochondria, neural stem cells, energy metabolism, aging

## Abstract

Mitochondria are crucial for cellular energy metabolism and are involved in signaling, aging, and cell death. They undergo dynamic changes through fusion and fission to adapt to different cellular states. In this study, we investigated the effect of knocking out the *dynamin 1-like protein* (*Dnm1l*) gene, a key regulator of mitochondrial fission, in neural stem cells (NSCs) differentiated from *Dnm1l* knockout embryonic stem cells (*Dnm1l*^−/−^ ESCs). *Dnm1l*^−/−^ ESC-derived NSCs (*Dnm1l*^−/−^ NSCs) exhibited similar morphology and NSC marker expression (Sox2, Nestin, and Pax6) to brain-derived NSCs, but lower *Nestin* and *Pax6* expression than both wild-type ESC-derived NSCs (WT-NSCs) and brain-derived NSCs. In addition, compared with WT-NSCs, *Dnm1l*^−/−^ NSCs exhibited distinct mitochondrial morphology and function, contained more elongated mitochondria, showed reduced mitochondrial respiratory capacity, and showed a metabolic shift toward glycolysis for ATP production. Notably, *Dnm1l*^−/−^ NSCs exhibited impaired self-renewal ability and accelerated cellular aging during prolonged culture, resulting in decreased proliferation and cell death. Furthermore, *Dnm1l*^−/−^ NSCs showed elevated levels of inflammation and cell stress markers, suggesting a connection between *Dnm1l* deficiency and premature aging in NSCs. Therefore, the compromised self-renewal ability and accelerated cellular aging of *Dnm1l*^−/−^ NSCs may be attributed to mitochondrial fission defects.

## 1. Introduction

Mitochondria are essential organelles at the core of cellular energy metabolism and play vital roles in signaling pathways, aging, and cell death [1,2]. They are highly plastic and dynamic, allowing them to rapidly adapt to diverse cellular environments and meet the biological demands of different cell types [3,4,5,6,7,8]. For example, pluripotent stem cells, such as embryonic stem cells (ESCs), typically contain globular mitochondria, whereas differentiated cells, including somatic stem cells, contain elongated mitochondria [3,4,5,6,7]. Furthermore, continuous structural remodeling, known as mitochondrial dynamics, within the mitochondrial network occurs through fusion and fission, which are essential for proper functioning in different cellular states [9]. Moreover, maintaining a balanced transition between mitochondrial fusion and fission is vital for ensuring cell adaptability and sufficient energy availability [8]. Numerous proteins are involved in mitochondrial fission and fusion. Among these regulatory proteins, dynamin 1-like protein (Dnm1l, known as Drp1 in humans) is directly involved in fission; it is recruited to the outer mitochondrial membrane and forms a ring-like structure that contracts the mitochondrial membrane [10,11]. Previous studies using knockout mice have highlighted the importance of *Dnm1l* for the proper functioning of organs such as the brain and heart [12,13,14]. In a recent study, we further demonstrated that *Dnm1l* knockout resulted in mitochondrial morphology and energy metabolism changes in mouse ESCs without the loss of pluripotency or differentiation potential in the three germ layers [15]. The fate of mitochondria is determined by the location of fission. When fission occurs at the periphery of mitochondria, it is degraded via mitophagy [16]. Therefore, normal fission of mitochondria by the Dnm1l protein is critical for maintaining mitochondrial function.

In neurogenesis, changes in mitochondrial morphology and function are critical regulatory factors that affect cell fate decisions at the beginning of neural development [17,18]. The initiation of neural stem cell (NSC) differentiation entails a shift in the major energy metabolism from glycolysis to oxidative phosphorylation (OXPHOS) [19,20,21]. During neuronal differentiation from NSCs, mitochondria undergo dynamic morphological changes. The elongated mitochondria in NSCs become fragmented in the neural progenitor cell (NPC) state and re-elongate once the cells differentiate into neurons [22,23]. Considering the role of mitochondria in neural development, it is reasonable to assume that defects in mitochondrial dynamic-related phenotypes could potentially lead to impaired neurogenesis [22,24]. Although many studies have shown a correlation between mitochondrial dysfunction and neurological disorders [25,26], the mechanisms that characterize mitochondrial dysfunction in NSCs are poorly understood.

Previously, we demonstrated, through in vitro differentiation and teratoma formation, that *Dnm1l*^−/−^ ESCs retain their ability to differentiate into the three germ layers [15]. However, we did not examine whether *Dnm1l*^−/−^ ESCs could differentiate into specific cell types and maintain the functional attributes of the resulting cells. In the present study, we assessed the differentiation potential of *Dnm1l*^−/−^ ESCs into homogenous NSC lines and investigated the characteristics of the *Dnm1l*^−/−^ ESC-derived NSCs (*Dnm1l*^−/−^ NSCs).

## 2. Results

### 2.1. Differentiation of Dnm1l^−/−^ ESCs to NSCs In Vitro

In a previous study, we generated *Dnm1l*^−/−^ ESCs to investigate the effects of *Dnm1l* deficiency on self-renewal and the dynamics of mitochondrial energy metabolism in ESCs [15]. Here, we explored whether somatic stem cells differentiated from *Dnm1l*^−/−^ ESCs displayed cell type-specific characteristics. For this purpose, we differentiated *Dnm1l*^−/−^ ESCs into NSCs (Figure 1A) because NSCs can be efficiently derived from ESCs, and the resulting ESC-derived NSCs display homogeneous morphology and marker expression, along with elongated self-renewal ability [27,28]. *Dnm1l*^−/−^ ESCs maintained an undifferentiated state and formed dome-like colonies when cultured in a medium containing fetal bovine serum (FBS) on feeder-layered dishes (Figure 1B). During the onset of differentiation, *Dnm1l*^−/−^ ESCs underwent a change in colony morphology from dome-shaped to plate-shaped by day 2 of differentiation (Figure 1B). After suspension culture for 5–7 days in the presence of basic fibroblast growth factor (bFGF) and epidermal growth factor (EGF), *Dnm1l*^−/−^ ESCs formed aggregates (Figure 1D). Aggregates resembling rosette-like structures were selected and transferred to gelatin-coated dishes in NSC expansion medium supplemented with bFGF and EGF. Neural lineage outgrowth spread from the seeded aggregates, exhibiting a branched morphology characteristic of NSCs (Figure 1E). Outgrowths were collected and subsequently passaged every 3–4 days to establish a homogenous NSC population (Figure 1F). Notably, the established *Dnm1l*^−/−^ NSCs were morphologically similar to brain-derived NSCs (Figure 1F,G).

### 2.2. Characterization of Dnm1l^−/−^ NSCs

We investigated whether the established *Dnm1l*^−/−^ NSCs possessed characteristic features of brain-derived NSCs. In this study, NSCs from passage 8 were used, and NSC marker expression was assessed using both immunocytochemistry and real-time quantitative reverse transcription PCR (RT-qPCR) and their tripotent differentiation ability. Immunocytochemistry revealed the expression of the NSC markers Nestin and Sox2 in *Dnm1l*^−/−^ NSCs as well as wild-type ESC-derived NSCs (WT-NSCs) (Figure 2A). However, RT-qPCR showed that the expression levels of NSC markers (Sox2, Nestin, and Pax6) in *Dnm1l*^−/−^ NSCs were lower than those in WT-NSCs, which exhibited levels similar to brain-derived NSCs (Figure 2B). To evaluate the tripotent differentiation potential, *Dnm1l*^−/−^ NSCs were differentiated by culturing for 10 days in the absence of bFGF and EGF in culture medium. The presence of astrocytes (GFAP + cells) and oligodendrocytes (O1 + cells) was observed in the differentiated cells (Figure 2C). However, early neuronal Tuj1 + cells were not detected even after 10 days of inducement (Figure 2C).

To verify whether undifferentiated *Dnm1l*^−/−^ NSCs constituted a homogenous population devoid of differentiated neural subtypes, we stained them for Tuj1, GFAP, and O1 neural subtype markers. Interestingly, Tuj1-expressing cells were detected in the early stages of NSCs at passage 8 (Figure 2D,E), whereas GFAP- and O1-expressing cells were not detected. These results suggest that *Dnm1l*^−/−^ NSCs retained stemness, but there was ongoing differentiation towards the neuronal lineage under NSC culture conditions. This indicates that the stemness-maintaining ability of NSCs is attenuated by *Dnm1l* deficiency.

### 2.3. Comparison of Mitochondrial Morphology and Energy Metabolism between Dnm1l^−/−^ NSCs and Wild-Type ESC-Derived NSCs (WT-NSCs)

Mitochondria are dynamic intracellular organelles that undergo fission and fusion in response to differentiation and cellular states [7,8]. Dnm11 plays a crucial role as a key regulator of mitochondrial fission [10,29]. *Dnm1l*^−/−^ NSCs displayed a different mitochondrial morphology than WT-NSCs, indicating impaired mitochondrial fission (Figure 3A). *Dnm1l*^−/−^ NSCs displayed numerous abnormally swollen mitochondria, and the mitochondrial shape was more elongated in these cells (Figure 3A). To analyze mitochondrial morphology, standard parameters, including the longest axis (c-max) and shortest axis (c-min) of the mitochondria, were determined (n = 76) (Figure 3B). In *Dnm1l*^−/−^ NSCs, the c-max values were higher, whereas the c-min values were lower than those in WT-NSCs (Figure 3B). When comparing mitochondrial globularity using the c-max/c-min measurement, *Dnm1l*^−/−^ NSCs showed values approximately two times higher than WT-NSCs, indicating a significantly higher prevalence of elongated mitochondria (Figure 3C). This observation reinforced that the absence of *Dnm1l* leads to impaired mitochondrial fission, resulting in the elongation and abnormal swelling of mitochondria in *Dnm1l*^−/−^ NSCs.

Morphological changes in the mitochondria are associated with cellular metabolism [7,8,30,31]. We hypothesized that the distinct mitochondrial shapes observed in *Dnm1l*^−/−^ NSCs would lead to metabolic differences compared with control NSCs. Measurement of the oxygen consumption rate (OCR), which represents mitochondrial respiratory capacity, using a Seahorse analyzer revealed that *Dnm1l*^−/−^ NSCs showed decreased overall OXPHOS capacity compared with WT-NSCs (Figure 3D). Additionally, *Dnm1l*^−/−^ NSCs exhibited reduced basal and maximal respiration rates (Figure 3E). Quantitative analysis of OXPHOS (measured using OCR) and glycolysis (measured using extracellular acidification rate [ECAR]) levels represented in 2D graphs showed that *Dnm1l*^−/−^ NSCs displayed lower OCR and higher ECAR values, indicating an OXPHOS-to-glycolytic shift in energy metabolism caused by *Dnm1l* knockout (Figure 3F). Furthermore, the contribution of mitochondria to ATP production was lower, whereas that from glycolysis was higher in *Dnm1l*^−/−^ NSCs than in WT-NSCs (Figure 3G). Collectively, *Dnm1l*^−/−^ NSCs exhibit structural abnormalities with elongated mitochondria, reduced OXPHOS capacity, and a metabolic shift toward glycolysis for ATP production. These findings suggest that the *Dnm1l* deficiency affected mitochondrial morphology, leading to altered cellular energy metabolism in the *Dnm1l*^−/−^ NSCs.

### 2.4. Impaired Self-Renewal and Accelerated Cellular Aging in Dnm1l^−/−^ NSCs

One of the distinct phenotypes observed in *Dnm1l*^−/−^ NSCs was their inability to self-renew and survive in long-term culture. Unlike WT-NSCs, *Dnm1l*^−/−^ NSCs showed reduced capacity to undergo self-renewal and eventually died under prolonged culture conditions (Figure 4A). The proliferation rate of *Dnm1l*^−/−^ NSCs gradually decreased after passage 16 and almost ceased around passage 22 (Figure 4A), indicating a significant decrease in the proliferation rate compared with WT-NSCs (Figure 4A). Compared with WT-NSCs, which exhibited a high proliferation rate in both knock-out groups, *Dnm1l*^−/−^ NSCs showed a noticeable reduction in proliferation. This difference became even more pronounced after passage 20, as *Dnm1l*^−/−^ NSCs struggled to maintain their proliferative capacity (Figure 4A). These results indicate that the *Dnm1l* deficiency significantly impairs the long-term proliferative potential of *Dnm1l*^−/−^ NSCs, further highlighting the critical role of *Dnm1l* in regulating cellular proliferation and self-renewal.

We hypothesized that the loss of the self-renewal ability of *Dnm1l* deficient NSCs could be attributed to cellular senescence or premature aging. To address this, we analyzed the markers associated with inflammation and endoplasmic reticulum (ER) stress, which are indicators of aging. RT-qPCR analysis revealed that *Dnm1l*^−/−^ NSCs (passage 18) showed an increased expression of senescence markers, including inflammation (*MMP13*, *IL-1R1*, *CK68*), ER stress (*CHOP*, *IRE1*, *BiP*), and cell cycle (*β-galactosidase*, *P16*, *P21*, and *P53*) compared with control NSCs (passage 20) (Figure 4B–D). Collectively, our data suggest that mitochondrial dysfunction resulting from *Dnm1l* knockout may contribute to the reduced proliferation of NSCs.

## 3. Discussion

In this study, we conducted a comprehensive analysis of the characteristics of NSCs derived from *Dnm1l*^−/−^ ESCs. Although *Dnm1l*^−/−^ NSCs predominantly maintained a homogenous NSC population, differences were observed in the expression levels of NSC markers (*Nestin* and *Pax6*), the presence of Tuj1^+^ cells in the NSC culture conditions, alterations in mitochondrial morphology and energy metabolism, and defective self-renewal ability. We found that the impaired self-renewal observed in *Dnm1l*^−/−^ NSCs could be attributed to premature aging or senescence, as evidenced by increased inflammation and cellular stress. These processes are influenced by *Dnm1l* deficiency and resulting mitochondrial defects.

The expression of NSC markers was lower in *Dnm1l*^−/−^ NSCs than in control NSCs (WT-NSCs and brain-derived NSCs). Although *Dnm1l*^−/−^ NSCs predominantly retained a homogenous NSC population, a subset of these cells expressed Tuj1, an early neuronal marker, even during early passages, suggesting the potential effect of mitochondrial abnormalities on NSC stemness and function in sustaining undifferentiated NSCs. Moreover, during long-term culture, *Dnm1l*^−/−^ NSCs lost their self-renewal ability, ultimately leading to cell death, which may be attributed to premature aging mechanisms. This finding is consistent with previous reports that suggest that mitochondria govern cellular energy metabolism and participate in signaling pathways, aging, and cell death [1,2]. Furthermore, the increase in inflammation- and cell stress-associated markers suggests a complex interplay between mitochondrial dynamics and aging-related processes. Further investigations are warranted to explore the underlying molecular mechanisms in detail and gain a comprehensive understanding of the role of *Dnm1l* in maintaining stem cell homeostasis.

In our previous study, we compared the three knockout ESC lines: *Fis1*^−/−^, *Mff*^−/−^, and *Dnm1l*^−/−^ ESCs. Notably, *Dnm1l*^−/−^ ESCs exhibited remarkable changes in mitochondrial morphology, proliferation rates, and reliance on OXPHOS for energy production [15]. When comparing differentiation potential, *Dnm1l*^−/−^ ESCs generated teratomas containing all three germ layer tissues, but the size of the teratomas was significantly smaller than that of the control and other knockout ESC lines (*Fis1*^−/−^ and *Mff*^−/−^ ESCs) [15]. This study extends our understanding of *Dnm1l* deficiency in NSCs, wherein phenotypic alterations similar to those observed in *Dnm1l*^−/−^ ESCs, including changes in mitochondrial morphology, metabolic patterns, and proliferation rates, were observed. Interestingly, *Dnm1l* knockout in NSCs resulted in premature aging and eventual cell death during prolonged culture, but *Dnm1l* knockout in pluripotent ESCs did not induce cell death and led to sustained proliferation, albeit at a reduced rate. This finding holds significance in clinical applications, as the dysregulation of mitochondrial dynamics can cause neurodegeneration, a pathological feature observed in several neurodegenerative disorders. Notably, in neurogenesis, mitochondrial remodeling that occurs after cell division in NSCs determines the fate of daughter cells towards either self-renewal or differentiation [22,23]. In addition, abnormal inhibition and activation of mitochondrial dynamics occur in the early stages of many diseases, particularly neurodegenerative disorders [32,33,34]. A dominant negative mutation in *Dnm1l* in humans leads to impaired brain development and neonatal mortality [35]. Mutations in *Opa1*, *Opa3*, and *Opa7*, which are crucial genes for regulating mitochondrial dynamics, have been suggested to be associated with autosomal dominant optic atrophy [36]. Excessive mitochondrial fragmentation, which disturbs typical mitochondrial dynamics, is observed in Huntington’s disease. This disruption ultimately results in diminished motility and compromised respiratory function [32,37].

Although *Dnm1l*^−/−^ NSCs had elongated mitochondria, they exhibited a lower OCR than WT-NSCs and relied more on glycolysis for energy production. We anticipated that, as demonstrated in ESCs, *Dnm1l*^−/−^ NSCs, which contain more elongated mitochondria, would be more dependent on OXPHOS than WT-NSCs [15]. Wang et al. have demonstrated that a reduction in *Dnm1l* levels in NSCs leads to mitochondrial elongation and an increase in oxidative phosphorylation (OXPHOS) [38], which may seem contradictory to our result. This is because knock-out of *Dnm1l* causes mitochondrial elongation, but the mitochondria may not be fully functional, due to the numerous swollen parts of the mitochondria. (Figure 3A). Defects in mitochondrial dynamics lead to depletion of the NSC population in vivo [22]. Therefore, it is evident that the knock-out of the *Dnm1l* gene causes the low expression of NSC markers. Further studies are required to reveal the role of mitochondrial defects in completing ESC differentiation of NSC-like properties. Moreover, in *Dnm1l*^−/−^ NSCs, premature aging may cause a failure to eliminate dysfunctional mitochondria. This parallels the situation in aged cells, where diminished autophagic capacity hinders the removal of malfunctioning components, consequently contributing to cellular function deterioration [39,40]. Mitochondrial fission contributes to the removal of defective components via mitophagy [5,41,42]. However, this process is less efficient in aging cells and compromises mitochondrial quality [43,44]. The abnormal mitochondrial ultrastructure in *Dnm1l*^−/−^ NSCs suggests that proper mitophagy is hindered by *Dnm1l* deficiency, resulting in a decrease in mitochondrial function.

In conclusion, our findings suggested that *Dnm1l* deficiency leads to morphological and functional alterations in NSC mitochondria. The observed impairment in self-renewal and cellular aging may be linked to mitochondrial dysfunction and altered cellular energy metabolism in *Dnm1l*^−/−^ NSCs. These results contribute to our understanding of the important role of *Dnm1l* in regulating NSC properties and highlight its potential implications in cellular aging and neurodevelopmental disorders.

## 4. Materials and Methods

### 4.1. Mouse ESC Culture

The mitochondrial fission-related gene, *Dnm1l*^−/−^, ESCs and wild type E14tg2a mouse embryonic stem cells (WT-ESCs) were obtained from our laboratory as described previously [28,45]. *Dnm1l*^−/−^ ESCs and WT-ESCs were maintained on mitomycin-C-treated mouse embryonic fibroblasts (MEF) in ESC medium. The ESC medium consisted of Dulbecco’s modified Eagle’s medium (DMEM) low glucose (Gibco, 11885-084 Billings, MT, USA) containing 15% FBS (Hyclone, SH30910.03, Logan, UT, USA), 1% penicillin/streptomycin/glutamine (P/S/G; Gibco, 10378-016), 0.1 mM non-essential amino acids (Gibco, 11140050), and 1mM β-mercaptoethanol (Gibco, 21985-023) supplemented with 10^3^ U/mL leukemia inhibitory factor (LIF; Merck, ESC1107, Rahway, NJ, USA). The ESCs were incubated at 37 °C and passaged when they reached 80% confluence in culture dishes.

### 4.2. NSC Differentiation and Culture

*Dnm1l*^−/−^ NSC and WT-NSC were differentiated following a previously reported protocol with slight modifications [45]. To remove the MEF population, the ESCs were incubated in gelatin-coated dishes for 1 h. After incubation, 1 × 10^5^ cells were suspended in 15% FBS and DMEM low glucose base media and seeded onto a 3.5 cm gelatin-coated dish. After 2–3 days, the cells were dissociated into single cells and cultured in NSC-inducing medium in a 6 cm petri dish for 3 days. The NSC-inducing medium consisted of N2B27 medium supplemented with 10 μg/mL of basic fibroblast growth factor (bFGF; R&D system, 233-FB-01M, Minneapolis, MN, USA) and epidermal growth factor (EGF; Gibco, PHG0311). After 3 days, the aggregates were attached to a gelatin-coated dish and cultured in NSC expansion medium, which consisted of DMEM-F12 (Gibco, 11320033), containing N2 supplement (Gibco, 17502-048), 1X P/S/G (Gibco, 10378-016), 7.5% bovine serum albumin Fraction V (BSA; Gibco, 15260037), 10 ng/mL EGF (Gibco, PHG0311), and 10 ng/mL bFGF. Cells were verified with RT-qPCR, and *Sox2*, *Pax6*, and *Nestin* positive cell groups were selected.

Characterized NSCs were maintained in NSC expansion medium and single-cell dissociation using trypsin was performed every 3–4 days for passaging. In addition, brain-derived NSC were obtained from our laboratory as previously reported [46] and maintained in the same condition as ESC-derived NSCs.

### 4.3. OCR Analysis

The OCR was measured using a Seahorse Bioscience XFp analyzer (Seahorse Bioscience, North Billerica, MA, USA). The day before analysis, *Dnm1l*^−/−^ NSCs and WT-NSCs (7 × 10^4^ cells per well) were cultured in XFp cell culture plates coated with 0.1% Matrigel (Corning, 356234, Corning, NY, USA) for overnight incubation. An hour before analysis, the culture medium was changed to an assay medium consisting of XF DMEM base media supplemented with glucose (17.5 mM), sodium pyruvate (0.5 mM), and L-glutamine (2.5 mM). The plates were then incubated in a non-CO_2_ incubator. Following the manufacturer’s instructions, the cells were supplemented with three chemicals (final concentrations: oligomycin-1.5 μM, FCCP-2 μM, and rotenone/antimycin A-0.5 μM) and analyzed according to the Agilent Seahorse XFp Analyzer guidelines. Experiments were performed with biologically duplicated samples.

### 4.4. ATP Production Rate Assay

ATP production was measured using a Seahorse Bioscience XFp analyzer (Seahorse Bioscience, North Billerica, MA, USA). Cell preparation and assay media were the same as those used in the aforementioned OCR analysis. Following the manufacturer’s protocol, the cells were supplemented with two chemicals (final concentrations: oligomycin-1.5 μM and rotenone/antimycin A-0.5 μM) and analyzed according to the Agilent Seahorse XFp Analyzer guidelines. Experiments were performed with biologically triplicated samples.

### 4.5. Immunocytochemistry

The cell culture medium was removed, and the cells were fixed with 4% paraformaldehyde for 30 min at 4 °C. After washing with PBS, the cells were treated with a PBS wash solution containing 0.3% Triton X-100 for 10 min at room temperature. Subsequently, the cells were blocked by incubating in PBS containing 3% bovine serum albumin for 30 min at room temperature. The cells were then incubated with primary antibodies diluted in the blocking solution for 16 h at 4 °C. The primary antibodies used were anti-Nestin (Nestin; 1:500, monoclonal, Millipore MAB353, Burlington, MA, USA), anti-Sox2 (Sox2; 1:500, polyclonal, Millipore AB5603), anti-GFAP (GFAP; polyclonal, 1:500, Abcam ab7260, Cambridge, UK), anti-O1 (O1; 1:500, monoclonal, Invitrogen 14-6506-82, Waltham, MA, USA), and anti-TUBB3 (Tuj1; 1:500, polyclonal, Abcam ab7260). The samples were then washed and fluorescent-labeled using secondary antibodies (Alexa Fluor 488 and 568). Finally, the cells were stained with Hoechst, diluted in the wash solution for 3 min at room temperature to visualize the cell nuclei, and washed. Each cell group was biologically triplicated.

### 4.6. RNA Isolation and RT-qPCR

Total RNA was isolated using TRIzol™ Reagent (Invitrogen, 15596026) following the manufacturer’s instructions. RNA (1 μg) was reverse transcribed to cDNA using SuperiorScript III RT Master Mix (Enzynomics, RT300S, Daejeon, Republic of Korea) following the manufacturer’s protocol. RT-qPCR was conducted using TOPreal™ qPCR 2X PreMIX (Enzynomics, RT500M) on a Roche LightCycler 5480 (Roche, Basel, Switzerland). The primers used for the RT-qPCR are listed in Table 1. Experiments were performed with triplicated samples.

### 4.7. Proliferation Rate Assay

WT-NSC and *Dnm1l*^−/−^ NSC (1 × 10^5^) were seeded in gelatin-coated 6-well plates and incubated for 24, 48, and 72 h. Then, cells were harvested and manually counted. All experiments were performed with triplicated samples.

### 4.8. Transmission Electron Microscopy (TEM)

For TEM analysis, NSCs were fixed in 4% paraformaldehyde and 2.5% glutaraldehyde in 0.1 M phosphate buffer (PB) for 24 h. After washing with 0.1 M PB, the samples were post-fixed in 0.1 M PB supplemented with 1% osmium tetroxide for 1 h. The samples were then serially dehydrated with graded ethanol concentrations (50–100%). For polymerization, samples were embedded in Epon 812 and incubated overnight at 60 °C. Ultrathin sections (60–70 nm-thick) were obtained using an ultramicrotome (Leica UC7 Ultramicrotome, Leica Microsystems, Wetzlar, Germany). The sectioned samples were collected on a grid and stained with uranyl acetate and lead citrate. Finally, the prepared samples were observed under transmission electron microscope (JEOL, JEM 1010, Tokyo, Japan) operating at 60 kV and imaged using a charge-coupled device camera (Gatan, SC1000, Pleasanton, CA, USA). The images from the TEM were then analyzed by the ImageJ 1.53 (NIH) software for measurement of the mitochondrial length.

### 4.9. Statistical Analysis

All experiments were performed with triplicated samples, except for the OCR analysis, which was performed in duplicate. Data are presented as means ± standard deviation (SD). The significance of differences was determined using an unpaired two-tailed Student’s *t*-test and one-way analysis of variance with Tukey’s Honestly Significant Difference post hoc test and Duncan’s multiple range test for multiple comparisons. Statistical significance was set at *p* < 0.05.

## Figures and Tables

**Figure 1 ijms-24-14291-f001:**
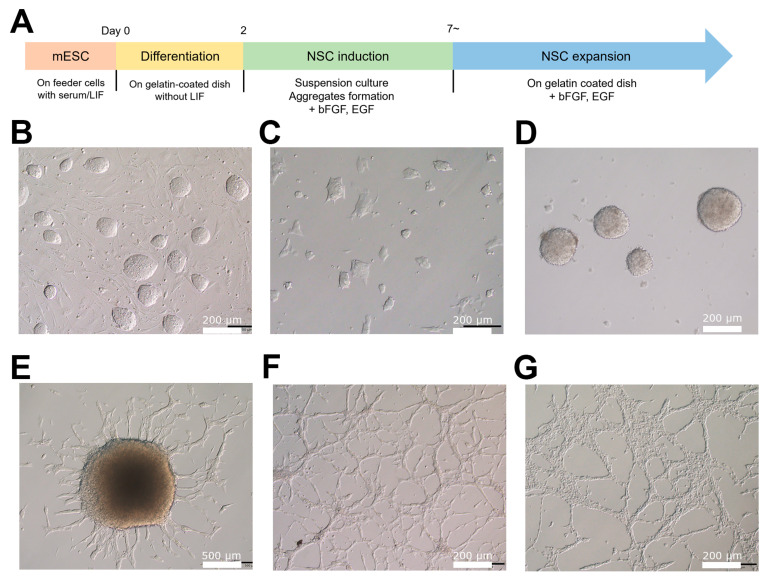
Differentiation of *Dnm1l* knockout embryonic stem cells (*Dnm1l*^−/−^ ESCs) into neural stem cells (NSCs). (**A**) Schematic illustration of the protocol for in vitro differentiation of mouse *Dnm1l*^−/−^ ESCs into NSCs. (**B**) Representative image of *Dnm1l*^−/−^ ESCs cultured on the feeder in the serum/leukemia inhibitory factor (LIF) condition. (**C**) Morphology of *Dnm1l*^−/−^ ESCs cultured for 2 days in 15% fetal bovine serum (FBS) medium without feeder and LIF supplementation. (**D**) Aggregates of differentiating *Dnm1l*^−/−^ ESCs forming embryonic bodies (EBs) in N2B27 suspension medium supplemented with basic fibroblast growth factor (bFGF) and epidermal growth factor (EGF) for 2 days. (**E**) Attached EBs forming outgrowths on the gelatin-coated dish. (**F**) Brightfield images of NSCs at passage 8 cultured in NSC expansion medium. (**G**) Established NSCs derived from the mouse brain.

**Figure 2 ijms-24-14291-f002:**
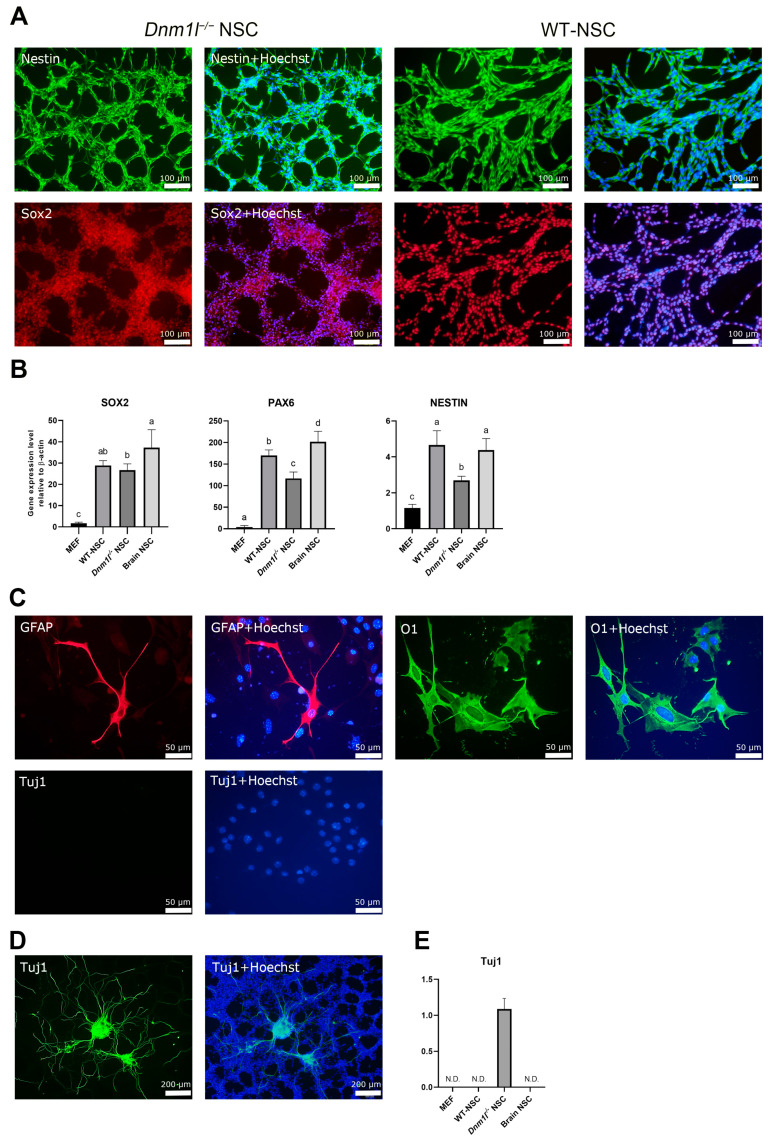
Characterization of NSCs differentiated from *Dnm1l*^−/−^ ESCs. (**A**) Immunocytochemistry analysis using NSC markers (Nestin and Sox2) in *Dnm1l*^−/−^ ESC-derived NSCs (*Dnm1l*^−/−^ NSCs) and wild-type ESC-derived NSCs (WT-NSCs). Nuclei were counterstained with Hoechst. (**B**) Real-time quantitative reverse transcription PCR (RT-qPCR) indicating the expression of NSC markers (*Sox2*, *Pax6*, and *Nestin*) in mouse embryonic fibroblasts (MEFs), WT-ESCs, *Dnm1l*^−/−^ NSCs, and mouse brain-derived NSCs (Brain NSCs). Different letters indicate significant differences among three cell types. (**C**) Immunocytochemistry analysis after in vitro differentiation of NSCs into neural lineage cells using GFAP (astrocyte marker), O1 (oligodendrocyte marker), and Tuj1 (early neuron marker). (**D**) Immunocytochemical analysis in *Dnm1l*^−/−^ NSCs using an early neuronal marker (Tuj1). Nuclei were counterstained with Hoechst. (**E**) RT-qPCR indicating the expression of *Tuj1* in MEFs, wild-type ESC-derived NSCs (WT-NSCs), *Dnm1l*^−/−^ NSCs, and Brain NSCs. All data are presented as the mean ± SD of three independent experiments. Statistical analysis was conducted using Duncan’s multiple-range test. N.D.: not detectable.

**Figure 3 ijms-24-14291-f003:**
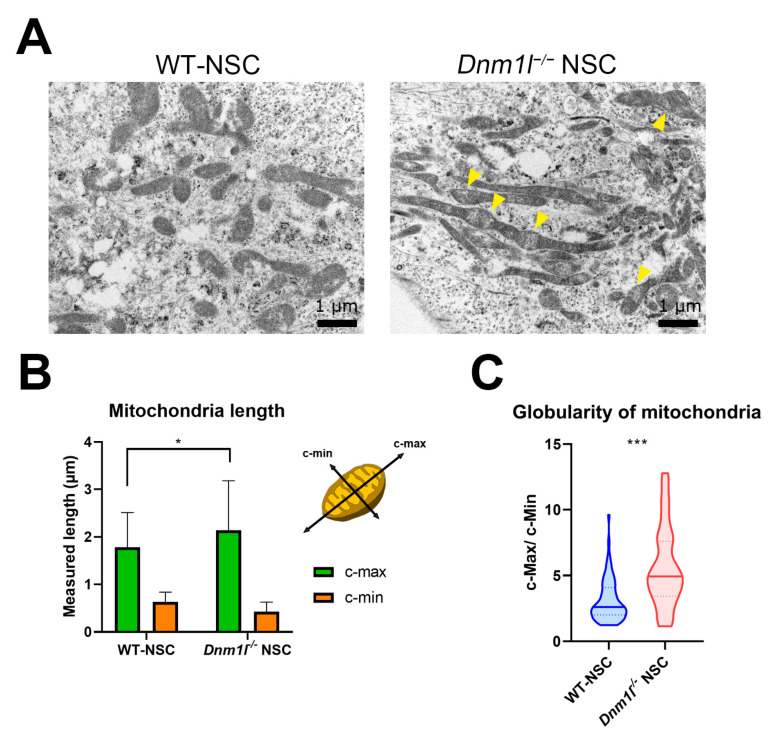
Ultrastructure and metabolic analysis of *Dnm1l*^−/−^ NSCs. (**A**) Representative transmission electron microscopy (TEM) images of mitochondria in WT- NSCs and *Dnm1l*^−/−^ NSCs. Mitochondria in *Dnm1l*^−/−^ NSCs were more elongated and displayed swollen mitochondria (yellow arrowheads). (**B**) Quantitively analyzed mitochondrial length in WT-NSCs and *Dnm1l*^−/−^ NSCs calculated based on the mitochondrial maximal (c-max) and minimal (c-min) axes using the ImageJ 1.53 software. * *p*-value < 0.01. (**C**) Globularity of mitochondria was analyzed by the ratio of two axes (c-max/c-min). *** *p*-value < 0.001. (**D**) The oxygen consumption rate (OCR) in WT-NSCs and *Dnm1l*^−/−^ NSCs. (**E**) Basal respiration and maximal respiratory capacity of WT-NSCs and *Dnm1l*^−/−^ NSCs. *** *p*-value < 0.001. (**F**) Measurement of energy map in WT-NSCs and *Dnm1l*^−/−^ NSCs. (**G**) Measurement of mitochondrial ATP and glycolytic ATP production in WT-NSCs and *Dnm1l*^−/−^ NSCs. All statistical analyses were conducted using Student’s *t*-test.

**Figure 4 ijms-24-14291-f004:**
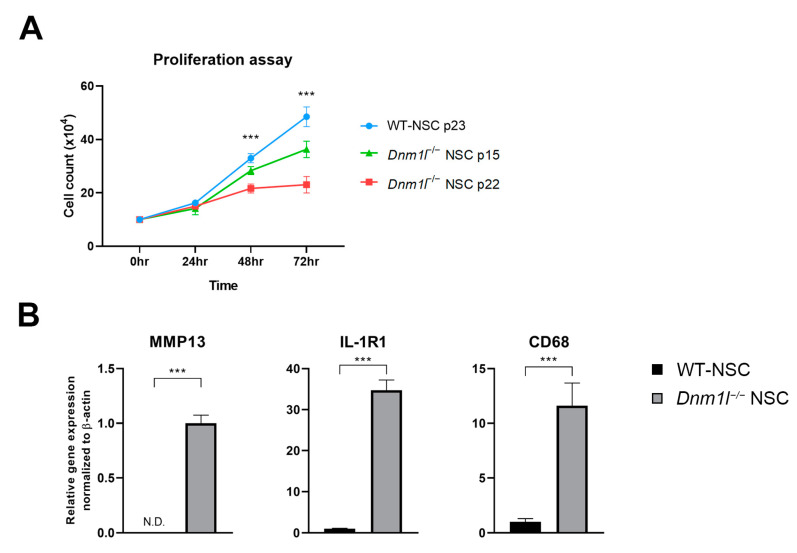
Proliferation rate and aging phenotype in WT-NSCs and *Dnm1l*^−/−^ NSCs. (**A**) Comparative proliferation rate analysis in WT-NSCs and *Dnm1l*^−/−^ NSCs. Statistical analyses were conducted using one-way ANOVA test. *** *p*-value < 0.001. (**B**) RT-qPCR showing the levels of inflammation markers (*MMP13*, *IL-1R1*, and *CD68*), (**C**) endoplasmic reticulum stress markers (*CHOP*, *IRE1*, and *BiP*), (**D**) and senescence markers (*β-galactosidase*, *P16*, *P21*, and *P53*). Data are expressed as the mean ± SD of three independent experiments. * *p*-value < 0.01, *** *p*-value < 0.001. Statistical analyses were conducted using Student’s *t*-test.

**Table 1 ijms-24-14291-t001:** Primers used for RT-qPCR.

Gene	Forward (5′ to 3′)	Reverse (5′ to 3′)
*Sox2*	GCG GAG TGG AAA CTT TTG TCC	CGG GAA GCG TGT ACT TAT CCT T
*Pax6*	CAG GTA TCC AAC GGT TGT G	GCT TAC AAC TTC TGG AGT CG
*Nestin*	AGA ACT CTC GCT TGC AGA C	GAG AAG GAT GTT GGG CTG A
*Tuj1*	GCT CAC GCA GCA GAT GTT CG	GGA TGT CAC ACA CGG CTA CC
*MMP13*	TGA TGA AAC CTG GAC AAG CA	GGT CCT TGG AGT GAT CCA GA
*IL-1R1*	GTG CTA CTG GGG CTC ATT TGT	GGA GTA AGA GGA CAC TTG CGA AT
*CD68*	GAA ATG TCA CAG TTC ACA CCA G	GGA TCT TGG ACT AGT AGC AGT G
*CHOP*	CCA CCA CAC CTG AAA GCA GAA	AGG TGA AAG GCA GGG ACT CA
*IRE1*	TTG AGA GAG CTT TTA CCA GCA G	ACC AGG ACC TGA CGG ATG T
*BiP*	TTC AGC CAA TTA TCA GCA AAC TCT	TTT TCT GAT GTA TCC TCT TCA CCA GT
*β-galactosidase*	GCA CGG CAT CTA TAA TGT CAC C	GTA TCG GAA TGG CTG TCC ATC
*P16*	CGC AGG TTC TTG GTC ACT GT	TGT TCA CGA AAG CCA GAG CG
*P21*	CCT GGT GAT GTC CGA CCT G	CCA TGA GCG CAT CGC AAT C
*P53*	GTC ACA GCA CAT GAC GGA GG	TCT TCC AGA TGC TCG GGA TAC
*β-actin*	CGC CAT GGA TGA CGA TAT CG	CGA AGC CGG CTT TGC ACA TG

## Data Availability

The data that support the findings of this study are available from the corresponding author upon reasonable request.

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
