# Peer review of "Altered Mitochondrial Function and Accelerated Aging Phenotype in Neural Stem Cells Derived from Dnm1l Knockout Embryonic Stem Cells"

_ijms, 2023, doi:10.3390/ijms241814291_

Round 1

Reviewer 1 Report

In this study, authors explored the role of Dnm1l, an important regulator involved in mitochondrial fission, in neural stem cells by using Dmn1l-knockout embryonic stem cells derived neural stem cells (Dnm1l(-/-)NSCs). Their findings showed that Dnm1l(-/-)NSCs are similar to brain derived NSCs in morphology and expression of NSC markers such as Sox2, Nestin, and Pax6. Interestingly, Nestin, and Pax6 expression in Dnm1l(-/-)NSCs were lower than WT-NSCs, and Dnm1l(-/-)NSCs had elongated mitochondria, reduced mitochondrial respiratory capacity, and a metabolic shift toward glycolysis for ATP production. Dnm1l(-/-)NSCs also showed other different features including impaired self-renewal, accelerated cellular aging, increased cell stress markers. These findings suggest that mitochondrial fission defects may compromise self-renewal ability and accelerate cell aging of NSCs. Overall, this study has interests and extends our knowledge to understand the impact of Dnm1l-mediated mitochondrial defects on the maintenance of stemness in neural stem cells. The experiments have been properly conducted, and most conclusions can be supported by the results. However, some issues still need to be addressed and several suggestions are listed to improve the present manuscript.

1. In fig.3, the findings show that ATP production rate is reduced in Dnm1l(-/-)NSCs. Is the total ATP content also decreased? In addition, is the number of mitochondria reduced in the Dnm1l(-/-)NSCs?

2. In fig.2 caption, the symbols (a, b, c, d) should be interpreted.

3. Authors observed that Nestin and Pax6 expression were decreased in Dnm1l(-/-)NSCs, Is there any link between downregulation of Nestin/Pax6 and mitochondrial defects?

4. In fig.4, authors hypothesized that cellular senescence or premature aging may be the reason why Dnm1l(-/-)NSCs lose self-renewal ability. However, important cellular senescence markers such as beta-galactosidase, p16, p21, p53, and gamma-H2A.X are not evaluated.

5. Wang et al report that Nestin loss triggers mitochondrial network remodeling and enhances oxidative phosphorylation (OXPHOS) and Nestin-Cdk5-Drp1 axis-reduced mitochondrial OXPHOS is indispensable for the maintenance of NSPC stemness (Stem cells 2018 36(4):589-601). On the contrary, authors observed that Dnm1l knockout reduced Nestin and OXPHOS in NSCs. Authors should further discuss these discrepancies.

Author Response

Response to Reviewers’ Comments

The authors thank the reviewer for the thoughtful comments. We have made substantial changes in the revised manuscript based on the reviewer’s suggestions. In the revised manuscript, we have highlighted the revised sections using red font for easy identification.

  1. In fig.3, the findings show that ATP production rate is reduced in Dnm1l(-/-)NSCs. Is the total ATP content also decreased? In addition, is the number of mitochondria reduced in the Dnm1l(-/-)NSCs?

à According to previous articles; (Chaphalkar et al., 2020; Cilenti et al., 2020), measurement of ATP production using metabolic analysis is accepted as a representative cellular respiration levels. Thus, it is reasonalble to view decrease of ATP production rate in Dnm1l-/- NSCs as evidence of decreased metabolic function. In addition, in our previous study (Seo et al., 2020), we found mtDNA copy number is lower in Dnm1l-/- cells than that of the wildtype group.

  1. In fig.2 caption, the symbols (a, b, c, d) should be interpreted.

à We have added the explanation for the symbols.

  1. Authors observed that Nestin and Pax6 expression were decreased in Dnm1l(-/-)NSCs, Is there any link between downregulation of Nestin/Pax6 and mitochondrial defects?

à Downregulation of NSC markers would be correlated with mitochondrial defects in way of differentiation due to metabolic malfunction. We have also added more related content to line 270 of the discussion.

Moreover, defects on mitochondrial dynamics lead to depletion of the NSC population in vivo (Khacho et al., 2016). Therefore, it is evident that the knock-out of Dnm1l gene causes the low expression of NSC markers. Further studies are required to reveal the role of mitochondrial defects in completing ESC differentiation of NSC-like properties.

  1. In fig.4, authors hypothesized that cellular senescence or premature aging may be the reason why Dnm1l(-/-)NSCs lose self-renewal ability. However, important cellular senescence markers such as beta-galactosidase, p16, p21, p53, and gamma-H2A.X are not evaluated.

à We agreed that cell cycle-related markers of aging needed to be identified. We have added additional data at Figure 4D with senescence markers, beta-galactosidase, p16, p21, p53.

  1. Wang et al report that Nestin loss triggers mitochondrial network remodeling and enhances oxidative phosphorylation (OXPHOS) and Nestin-Cdk5-Drp1 axis-reduced mitochondrial OXPHOS is indispensable for the maintenance of NSPC stemness (Stem cells 2018 36(4):589-601). On the contrary, authors observed that Dnm1l knockout reduced Nestin and OXPHOS in NSCs. Authors should further discuss these discrepancies.

à Thank you for your insights on the topic. We have added following explanation in the discussion, line 265.

Chaphalkar, R. M., Stankowska, D. L., He, S., Kodati, B., Phillips, N., Prah, J., . . . Krishnamoorthy, R. R. (2020). Endothelin-1 Mediated Decrease in Mitochondrial Gene Expression and Bioenergetics Contribute to Neurodegeneration of Retinal Ganglion Cells. Sci Rep, 10(1), 3571. doi:10.1038/s41598-020-60558-6

Cilenti, L., Di Gregorio, J., Ambivero, C. T., Andl, T., Liao, R., & Zervos, A. S. (2020). Mitochondrial MUL1 E3 ubiquitin ligase regulates Hypoxia Inducible Factor (HIF-1α) and metabolic reprogramming by modulating the UBXN7 cofactor protein. Sci Rep, 10(1), 1609. doi:10.1038/s41598-020-58484-8

Khacho, M., Clark, A., Svoboda, D. S., Azzi, J., MacLaurin, J. G., Meghaizel, C., . . . Slack, R. S. (2016). Mitochondrial Dynamics Impacts Stem Cell Identity and Fate Decisions by Regulating a Nuclear Transcriptional Program. Cell Stem Cell, 19(2), 232-247. doi:10.1016/j.stem.2016.04.015

Seo, B. J., Choi, J., La, H., Habib, O., Choi, Y., Hong, K., & Do, J. T. (2020). Role of mitochondrial fission-related genes in mitochondrial morphology and energy metabolism in mouse embryonic stem cells. Redox Biol, 36, 101599. doi:10.1016/j.redox.2020.101599

Reviewer 2 Report

In this study, Na et al., investigated the characteristics of NSCs derived from Dnm1l-/- ESCs. The authors claimed that Dnm1l-/- NSCs predominantly maintained a homogenous cell population, even though differences were observed for some NSC markers such as, Nestin and Pax6, for the presence of Tuj1+ cells, and for alterations in mitochondrial morphology and energy metabolism. The authors also found that Dnm1l-/- NSCs cells had impaired self-renewal capacity caused by premature aging (or senescence), as shown by increased inflammation and cellular stress.

The study might be interesting but it is weak and lacks several experiments to support the conclusions. The experimental setting is fuzzy, sometimes does not sound scientific and controls are often missing.

The authors presented limited examination of mitochondria defects in Dnm1l-/- ESCs. Metabolomics analysis of wild type and Dnm1l-/- NSCs would be required, as well as the analysis of mitochondria biogenesis, mtDNA copy number, mitophagy, oxidative stress, senescence and/or apoptosis. Have the authors tried to rescue the phenotype of knock-out cells by expressing wild type Dnm1l? How were Dnm1l-/- ESCs cells generated and characterized? Is a stable cell line?

Minor comments:

I would suggest to move “Material and Methods” before the “Results”.

-       “Material and Methods” lacks a lot of information (e.g., cells used, biological replicates, times an experiment has been repeated, how the absorbance was measured and analyzed…etc.).

For example, in lines 361-362 the authors stated that “All experiments were performed in triplicate, except for the OCR analysis, which was performed in duplicate” but this is confusing since it is not clear if the authors are referring to the number of biological replicates or to the times the experiment has been repeated.

-       Pag. 2, lines 48-50: Here, there is a scientific inaccuracy. The authors should know that fission may occur in the absence of mitochondrial DNA (mtDNA) replication, and the copies of mtDNA are simply “divided” in the new mitochondria. However, fission cannot be confused with mitochondrial biogenesis that, in contrast, involves complete mtDNA replication.

-       Pag. 2, lines 51-52: This statement is incorrect. Mitochondrial fission and mitochondrial biogenesis are distinct processes. Fission plays an important role in the removal of damaged mitochondria by mitophagy, and with fusion contributes to the maintenance of mitochondrial function and optimize bioenergetic capacity.

-       Pag. 2, line 64: It would be more appropriate to say that the mechanism(s) that characterize mitochondrial dysfunction in NSCs are poorly understood.

-       Pag. 3, section 2.2: Where are the controls (e.g., wild-type ESC-derived NSCs and/or brain-derived NSCs) for immunohistochemistry? This information should be provided due to RT-qPCR results.

-       Pag. 3, line 114.  In which experimental setting Tuj1+ cells were not detected after 10 days differentiation? What is reported in Fig.2 D? Controls are missing in this experiment.

-       Pag. 5, lines 131-132: The authors should keep consistency when describing the results. Sometimes they use cell passage number and other times days of differentiation. This is confusing.

-       Pag. 5, line 144: How many EM micrographs have been acquired? At which magnification? I suppose that n=76 represents the number of mitochondria included in the analysis. How did the authors do the EM measurements/analysis? Since mitochondrial fusion is primarily controlled by three GTPases, mitofusin 1 (Mfn1), Mfn2, and optic atrophy 1 (Opa1), to confirm the EM results, have the authors investigated for example the expression of these proteins as well the “down-regulation” of the GTPase dynamin-related protein 1?

-       What is the difference between Fig. 4 A and 4 B? In Fig. 4 A: How did the authors measure proliferation? In this Figure cell viability is drastically decreased after 72 hrs. How did the authors keep the cells in culture till passage 22? Why did the authors compare wt cells passage 23 and knock-out cells passage 22? This does not sound appropriate.

Because, after passage 16 there is a drastic reduction in cell viability (induction of apoptosis?) the authors should have been using lower cell passage number.

-       Pag. 10, line 280: Are MEFs maintained in mitomycin-C as feeder cells for ESCs? Please, specify it because it is not clear.

-       Pag. 10, line 285: What are mESCs?

-       Pag. 10, line 288: please, provide the reference(s).

-       Pag. 10, section 4.1: How the cells were kept? Incubator 37C?

-       Pag. 10, section 4.2: Did the authors check the expression of specific cell differentiation markers? Please, specify it.

-       Pag. 10, section 4.3: Many information is missing. For example, how many cell types were investigated? How many biological replicates? How many times the experiment has been performed?... etc. For example, pag. 3, lines 108-109 the authors introduced Dnm1l-/- NSCs and wild-type ESC-derived NSCs (WT-ESCs) that are not described under “Material and Methods”.

-       Pag. 10, section 4.4: Same comment, as above for section 4.3.

-       Pag. 10, section 4.5: Please, specify the cells used in the study and include how many times the experiment has been repeated.

Minor English editing is required.

Author Response

Response to Reviewers’ Comments

The authors thank the reviewer for the thoughtful comments. We have made substantial changes in the revised manuscript based on the reviewer’s suggestions. In the revised manuscript, we have highlighted the revised sections using red font for easy identification.

The authors presented limited examination of mitochondria defects in Dnm1l-/- ESCs. Metabolomics analysis of wild type and Dnm1l-/- NSCs would be required, as well as the analysis of mitochondria biogenesis, mtDNA copy number, mitophagy, oxidative stress, senescence and/or apoptosis. Have the authors tried to rescue the phenotype of knock-out cells by expressing wild type Dnm1l? How were Dnm1l-/- ESCs cells generated and characterized? Is a stable cell line?

à In our previous study (Seo et al., 2020), we successfully established and characterized several stable Dnm1l knock-out ESC lines. In that paper, we also observed mitochondrial morphology, energy metabolism, and gene expression levels in Dnm1l-/- ESCs. This study sought to investigate the Dnm1l knock-out effect to the somatic cells other than pluripotent ESCs. Our results are meaningful, because abnormalities in mitochondrial morphology and function were examined in Dnm1l-/- NSCs as well. We also found that Dnm1l-/- NSCs suffer from premature aging, which was not observed in Dnm1l-/- ESCs.

For recovery of the KO gene, we had experimented on Dnm1l expression recovery by introducing lentiviral vector to EF1a binding site (which is commonly used as safe harboring). With confirmation of gene insertion, we evaluated Dnm1l gene and protein expression level. However, our results were shown to be negative. Although could not put the recovery data in the manuscript, our data on Dnm1l-/- ESCs were triplicated with multiple colleagues, and we are confident our presented data is accurate.

Minor comments:

I would suggest to move “Material and Methods” before the “Results”.

à In the format of “International Journal of Molecular Sciences” journal, “Results” section is followed by “Material and Methods” section.

 “Material and Methods” lacks a lot of information (e.g., cells used, biological replicates, times an experiment has been repeated, how the absorbance was measured and analyzed…etc.).

à Further information has been added in “Material and Methods” section.

For example, in lines 361-362 the authors stated that “All experiments were performed in triplicate, except for the OCR analysis, which was performed in duplicate” but this is confusing since it is not clear if the authors are referring to the number of biological replicates or to the times the experiment has been repeated.

àExperiments were performed with triplicated biological replicates. The phrase was added in the main text.

Pag. 2, lines 48-50: Here, there is a scientific inaccuracy. The authors should know that fission may occur in the absence of mitochondrial DNA (mtDNA) replication, and the copies of mtDNA are simply “divided” in the new mitochondria. However, fission cannot be confused with mitochondrial biogenesis that, in contrast, involves complete mtDNA replication.

à According to your comments, the incorrect and unnecessary sentences have been revised.

Pag. 2, lines 51-52: This statement is incorrect. Mitochondrial fission and mitochondrial biogenesis are distinct processes. Fission plays an important role in the removal of damaged mitochondria by mitophagy, and with fusion contributes to the maintenance of mitochondrial function and optimize bioenergetic capacity.

Pag. 2, line 64: It would be more appropriate to say that the mechanism(s) that characterize mitochondrial dysfunction in NSCs are poorly understood.

à According to your comments, the incorrect and unnecessary sentences have been revised.

Pag. 3, section 2.2: Where are the controls (e.g., wild-type ESC-derived NSCs and/or brain-derived NSCs) for immunohistochemistry? This information should be provided due to RT-qPCR results.

à Data using wild-type NSCs have been added in Fig. 2A.

Pag. 3, line 114.  In which experimental setting Tuj1+ cells were not detected after 10 days differentiation? What is reported in Fig.2 D? Controls are missing in this experiment.

à After day 10 of neural lineage induction with Dnm1l-/- NSC, Tuj1+ cells were disappeared. Fig. 2D has been modified with additional data.

Pag. 5, lines 131-132: The authors should keep consistency when describing the results. Sometimes they use cell passage number and other times days of differentiation. This is confusing.

à No, passage number and “day of differentiation” cannot be kept consistent. Please understand this.

Pag. 5, line 144: How many EM micrographs have been acquired? At which magnification? I suppose that n=76 represents the number of mitochondria included in the analysis. How did the authors do the EM measurements/analysis?

à We acquired 14-15 micrographs at 5000X and 20000X magnification for each sample. We measured the length of mitochondria using ImageJ. These sentences has been added in the main text.

Since mitochondrial fusion is primarily controlled by three GTPases, mitofusin 1 (Mfn1), Mfn2, and optic atrophy 1 (Opa1), to confirm the EM results, have the authors investigated for example the expression of these proteins as well the “down-regulation” of the GTPase dynamin-related protein 1?

à Dnm1l (GTPase dynamin-related protein 1) is a mitochondrial fission-related gene. Thus, other fission-related genes, such as Fis1 and Mff, may be related to Dnm1l rather than Mfn1, Mfn2, and Opa1. In previously published data (Seo BJ et al., 2020. Redox Biology), we found that only Dnm1l-/- ESCs showed defective mitochondrial fission that follow the elongation of mitochondria, which was not observed in Fis1-/- and Mff-/- ESCs. This is why we use Dnm1l-/- ESCs rather than Fis1-/- and Mff-/- ESCs for further experiments.

What is the difference between Fig. 4 A and 4 B? In Fig. 4 A: How did the authors measure proliferation?

àAs you pointed out, Figure 4A and 4B did not show significant differences. So we have delited Fig. 4B, and “Proliferation rate assay” has been added to “Meterial and Method” section.

In this Figure cell viability is drastically decreased after 72 hrs. How did the authors keep the cells in culture till passage 22?

à In Fig. 4A, 72 hrs refers to the culture time after passage 22.

Why did the authors compare wt cells passage 23 and knock-out cells passage 22? This does not sound appropriate. Because, after passage 16 there is a drastic reduction in cell viability (induction of apoptosis?) the authors should have been using lower cell passage number.

à We sought to highlight the differences in proliferation rate between WT-NSCs and Dnm1l-/- NSCs. Dnm1l-/- NSCs ceased proliferation at p22. However, even later passage (p23) of WT-NSCs still proliferate well. Fig 4A is complemented with proliferate assay of passage 15 cells.

Pag. 10, line 280: Are MEFs maintained in mitomycin-C as feeder cells for ESCs? Please, specify it because it is not clear.

à As feeder cells for supporting ESCs, MEFs should be treated mitomycin C. This is a common process in ESC culture. Of course mitocycin C-treated FEMs do not proliferate but maintained for more than 2 weeks.

Pag. 10, line 285: What are mESCs?

à mESCs referred to mouse ESCs. We have revised the word.

Pag. 10, line 288: please, provide the reference(s).

à We have added the reference.

Pag. 10, section 4.1: How the cells were kept? Incubator 37C?

à We have added more information for cell culture condition.

Pag. 10, section 4.2: Did the authors check the expression of specific cell differentiation markers? Please, specify it.

à We have added differentiation markers. This was verified in section 4.2.

Pag. 10, section 4.3: Many information is missing. For example, how many cell types were investigated? How many biological replicates? How many times the experiment has been performed?... etc. For example, pag. 3, lines 108-109 the authors introduced Dnm1l-/- NSCs and wild-type ESC-derived NSCs (WT-ESCs) that are not described under “Material and Methods”.

à We have added more information in “Material and Methods” section.

Pag. 10, section 4.4: Same comment, as above for section 4.3.

à We have supplemented more information in “Material and Methods” section.

Pag. 10, section 4.5: Please, specify the cells used in the study and include how many times the experiment has been repeated.

à We have added the information about the time of the experimets and specific cell types used in this study.  

Seo, B. J., Choi, J., La, H., Habib, O., Choi, Y., Hong, K., & Do, J. T. (2020). Role of mitochondrial fission-related genes in mitochondrial morphology and energy metabolism in mouse embryonic stem cells. Redox Biol, 36, 101599. doi:10.1016/j.redox.2020.101599

Round 2

Reviewer 1 Report

Most of the previous issues have been resolved and the manuscript has been revised appropriately. However, the primers used in the newly added RT-qPCR results in fig. 4D were missing. Authors should add the sequences of these primers in table 1.

Author Response

Most of the previous issues have been resolved and the manuscript has been revised appropriately. However, the primers used in the newly added RT-qPCR results in fig. 4D were missing. Authors should add the sequences of these primers in table 1.

→ Newly added primers have been added in table 1.

Reviewer 2 Report

I appreciate the efforts of the authors to revise the manuscript based on my comments. However, the authors have not addressed all the concerns previously raised. I still find scientific inaccuracies. In the text, the authors justify some of their "weak" results referring to previous published work. This is not scientific, since each work should have a clear study design and control settings. In the future, I strongly recommend the authors to consider more carefully their experimental setting and carefully look at the consistency of their data.

Minor English editing is required

Author Response

I appreciate the efforts of the authors to revise the manuscript based on my comments. However, the authors have not addressed all the concerns previously raised. I still find scientific inaccuracies. In the text, the authors justify some of their "weak" results referring to previous published work. This is not scientific, since each work should have a clear study design and control settings. In the future, I strongly recommend the authors to consider more carefully their experimental setting and carefully look at the consistency of their data.

Minor English editing is required

→ Thank you for your understanding and effort to enhance the quality of our article. This manuscript had already undergone professional English editing before the initial submission. We hereby attached the Certificate.
